# Wind farms providing secondary frequency regulation: Evaluating the performance of model-based receding horizon control*

Carl R. Shapiro[1], Johan Meyers[2], Charles Meneveau[1], and Dennice F. Gayme[1]

[1]Department of Mechanical Engineering, Johns Hopkins University, 3400 N Charles St, Baltimore, Maryland 21218, USA
[2]Department of Mechanical Engineering, KU Leuven, Celestijnenlaan 300A, 3001 Leuven, Belgium

*Correspondence to:* Dennice Gayme (dennice@jhu.edu)

**Abstract.** We investigate the use of wind farms to provide secondary frequency regulation for a power grid using a model-based receding horizon control framework. In order to enable real-time implementation, the control actions are computed based on a time-varying one-dimensional wake model. This model describes wake advection and wake interactions, both of which play an important role in wind farm power production. In order to test the control strategy, it is implemented in a large eddy simulation (LES) model of an 84-turbine wind farm using the actuator disk turbine representation. Rotor-averaged velocity measurements at each turbine are used to provide feedback for error correction. The importance of including the dynamics of wake advection in the underlying wake model is tested by comparing the performance of this dynamic-model control approach to a comparable static-model control approach that relies on a modified Jensen model. We compare the performance of both control approaches using two types of regulation signals, "RegA" and "RegD", which are used by PJM, an independent system operator in the Eastern United States. The poor performance of the static-model control relative to the dynamic-model control demonstrates that modeling the dynamics of wake advection is key to providing the proposed type of model-based coordinated control of large wind farms. We further explore the performance of the dynamic-model control via composite performance scores used by PJM to qualify plants for regulation. Our results demonstrate that the dynamic-model controlled wind farm consistently performs well, passing the qualification threshold for all fast-acting RegD signals. For the RegA signal, which changes over slower time scales, the dynamic-model control leads to average performance that surpasses the qualification threshold, but further work is needed to enable this controlled wind farm to achieve qualifying performance for all regulation signals.

## 1 Introduction

Recent market trends are rapidly changing the composition of power grid energy sources, replacing conventional dispatchable power sources with non-dispatchable, variable resources, such as wind energy. These changes are putting pressure on the power system by reducing the number of resources available to provide a wide range of grid services traditionally provided by conventional power plants (Aho et al., 2012). A particularly important example is grid frequency regulation, which is closely tied to short-term imbalances in active power generation and load over time scales ranging from milliseconds to tens

---

*This paper is an extended version of our paper presented at the 2016 TORQUE conference: Shapiro, C. R., Meyers, J., Meneveau, C., and Gayme, D. F.: Wind farms providing secondary frequency regulation: Evaluating the performance of model-based receding horizon control, Journal of Physics: Conference Series, 753, 052012, 2016.

of minutes (Rebours et al., 2007). In order to deal with this challenge, a number of independent system operators (ISOs) are beginning to consider requiring wind plants to provide frequency regulation services and expanding frequency regulation markets to include wind plants (Aho et al., 2012; Díaz-González et al., 2014).

Secondary frequency regulation, where participating generators track a power signal sent by an ISO over tens of minutes, is an area of growing interest. Recent work (Aho et al., 2013; Jeong et al., 2014) has shown that stand-alone wind turbines can effectively provide secondary frequency regulation, but recent fluid dynamics simulations (Fleming et al., 2016) have shown that interactions between wakes can lead to poor tracking performance when these single turbine control strategies are applied to an array of turbines (Aho et al., 2013; Jeong et al., 2014). This poor performance is not unexpected because aerodynamic interactions between turbines occur at timescales commensurate with those of the secondary frequency regulation signals. Such considerations have led to recent emphasis on approaches that consider the collective behavior of the farm (Annoni et al., 2016; Gebraad et al., 2015), but few of these studies provide real-time implementable algorithms that can respond to changing wind farm power output levels.

Our recent work (Shapiro et al., 2017a) sought to overcome these challenges by developing a time-varying extension of the classic Jensen wake model (Katić et al., 1986) that accounts for the dynamics of wake advection through the farm. This new wake model was incorporated into a predictive model-based receding horizon control framework to coordinate an array of wind turbines to provide secondary frequency regulation by modulating the thrust coefficients of individual turbines. This approach used predictions from the underlying model to iteratively solve an online optimization problem representing the power tracking goal. Feedback from measurements of the velocity at each turbine was used to correct modeling errors. This approach showed promising results when tested in a large eddy simulation (LES) model of a wind farm where turbines were represented using actuator disk models (Shapiro et al., 2017a). In these simulations, we used setpoint reductions of only 50% of the maximum regulation provided, but were able track a sample regulation signal with the wind farm test system used. In previous studies (Aho et al., 2013; Jeong et al., 2014), where control design was done at the individual turbine level, successful power tracking required setpoint reductions exactly equal to the maximum change in power production requested by the ISO. The ability to lower the setpoint reduction represents an important advantage over single turbine approaches as the amount of setpoint reduction corresponds directly to the amount of power that wind farms are sacrificing in the bulk energy market to provide regulation. In fact, previous studies have shown that setpoint reductions equal to the one-sided regulation signal variation may not be economically prudent (Rose and Apt, 2014).

The feasibility of providing secondary frequency regulation with wind farms was demonstrated by our initial results (Shapiro et al., 2017a). In this work we further evaluate the performance of this approach and consider the effect of reducing the control design and wake model complexity. In particular, we evaluate the importance of explicitly modeling the dynamics of wake advection by comparing the performance of the dynamic-model approach to a similar static-model approach; i.e. we replace the dynamic wake model with a wake model that does not include wake advection (Katić et al., 1986). In order to make appropriate comparisons, the static-model controller solves an online optimization problem with feedback similar to that solved in the dynamic-model controller.

Both controllers are implemented in LES with actuator disk turbine models, which is used as a "virtual wind farm". We evaluate the two approaches with regulation test signals from PJM, an ISO in the United States Eastern Interconnection (PJM, 2012, 2015). PJM has two types of secondary frequency regulation signals that are based on the Area Control Error (ACE) signal, a combined measure of the power imbalance and deviation of the frequency from its nominal operating value. The "RegA" signal is a low pass filter of the ACE that is generally followed using traditional regulating resources, such as fossil fuel plants. The "RegD" signal is a high pass filter of the ACE that can be followed by more quickly responding resources, such as energy storage devices. Our results show that the static wake model leads to poor tracking performance, which indicates that the complexity of this particular control design cannot be reduced by ignoring the dynamics of wake advection. We then evaluate the performance of the dynamic-model controller using PJM's performance evaluation criteria (PJM, 2012, 2015) to determine whether the controlled wind farm system can meet PJM's threshold for qualification in the two regulation markets. These computations allow us to evaluate whether wind farms with this control strategy are better suited to provide traditional or fast-acting regulation.

The remainder of this paper is organized as follows. The static and dynamic wake models are described in Section 2, and the respective model-based controller designs are outlined in Section 3. Sections 4 and 5 describe the virtual wind farm test system and the simulation cases. The two controllers are compared in Section 6. The performance of dynamic-model control is further explored in Section 7 using PJM's performance criteria. Finally, we present conclusions and discuss directions for future work in Section 8.

## 2 Wake models

The two wake models employed here are respectively based on static and dynamic adaptations of the classic Jensen wake model (Katić et al., 1986). In this presentation of the Jensen model, we consider regularly arranged wind farms with $N$ rows and $M$ columns of turbines, where each column is aligned with the prevailing wind direction, as shown in Figure 1. The streamwise coordinate is denoted as $x$, and the $n$-th turbine row is located at $x = s_n$. Every turbine is assumed to have the same rotor diameter $D$.

The standard Jensen model assumes each turbine generates a wake region that expands radially at a linear rate $k$ with increasing downstream distance from the turbine. This leads to following definition of the wake diameter

$$D_w(x) = D + 2kx, \tag{1}$$

where $x$ is the streamwise distance from the turbine rotor plane. Conservation of mass leads to the following velocity deficit in the wake of the $m$-th turbine in the $n$-th row

$$\delta u_{nm}(x) = \frac{2U_\infty a_{nm}}{[D_w(x - s_n)/D]^2}, \tag{2}$$

where $a_{nm}$ is the induction factor and $U_\infty$ is the velocity upstream of the wind farm. This representation yields top-hat profiles of velocity deficits in each cross-stream plane. The velocity field experienced by the each turbine is found by superimposing

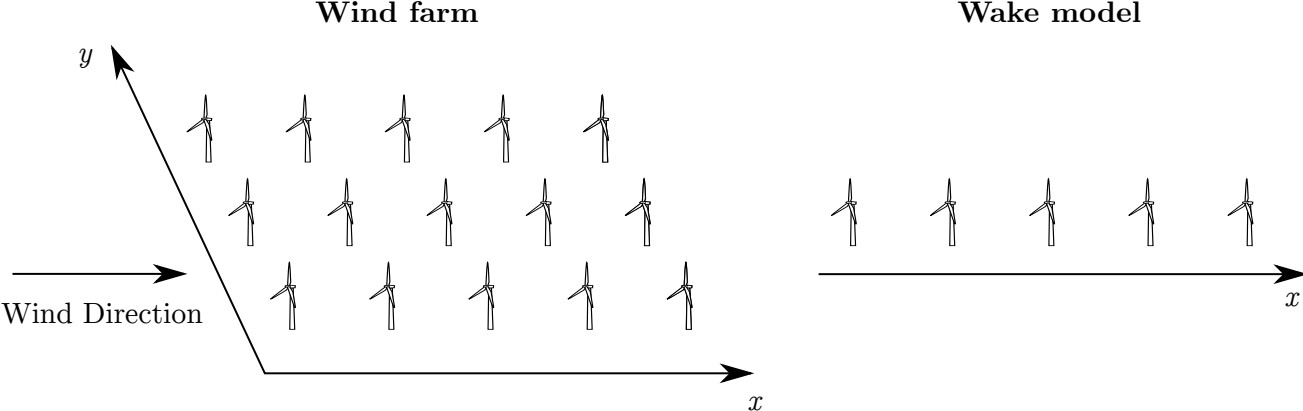

**Figure 1.** Diagram of a regular wind farm (left) with $N = 5$ rows and $M = 3$ columns of turbines and the corresponding row-averaged wake model representation (right). Each column is aligned with the streamwise coordinate $x$, and each row is aligned with the spanwise coordinate $y$.

the squared velocity deficits

$$u_{\infty nm} = U_\infty - \sqrt{\sum_{(j,k) \in \mathcal{S}_{nm}} \delta u_{jk}^2 (s_n - s_j)}, \tag{3}$$

where $\mathcal{S}_{nm}$ is defined as the set of turbines whose wakes lie within the swept area of the turbine rotor of the $m$-th turbine in row $n$. The definition of these sets means that Eq. (3) reduces to $u_{\infty 1m} = U_\infty$ for the first row of turbines. The power production
of each turbine is subsequently found using

$$\hat{P}_{nm} = \frac{1}{2} \rho \frac{\pi D^2}{4} C_{Pnm} u_{\infty nm}^3, \tag{4}$$

where $C_{Pnm}$ is the power coefficient of the turbine in row $n$ and column $m$.

For ease of implementation, each wake model used in this paper makes the following modifications to the standard Jensen model. First, we consider each row of turbines collectively (as shown in Figure 1; cf. Shapiro et al., 2017a), which means that
each modeled value is homogeneous in the spanwise direction and we neglect the spanwise merging of wakes. To reflect this modification, the column index $m$ used in the Eqs. (2)–(3) is dropped in subsequent equations. Second, to account for entrance effects in the farm and compensate for the neglected spanwise wake interactions, we allow each wind turbine row to have a separate wake expansion rate $k_n$.

Furthermore, we express the turbine power production using the local thrust coefficient $C'_{Tn}$ and modeled velocity at the
turbine rotor $\hat{u}_n$. Simple momentum theory can be used to show that (Meyers and Meneveau, 2010; Goit and Meyers, 2015; Shapiro et al., 2017a)

$$a_n = \frac{C'_{Tn}}{4 + C'_{Tn}}, \qquad \hat{u}_n = (1 - a_n) u_{\infty n}, \quad \text{and} \qquad C'_{Tn} = \frac{C_{Tn}}{(1 - a_n)^2}. \tag{5}$$

Similarly, one can show that $C'_{Pn} = C_P/(1-a_n)^3$, from which we conclude $C'_{Tn} = C'_{Pn}$. Replacing the induction factor $a_n$ in Eq. (2), the modeled upstream velocity $u_{\infty n}$ in Eq. (3), and the power power coefficient $C_{Pn}$ in Eq. (4) with these equations, the power production can be rewritten as

$$\hat{P}_n = M\frac{1}{2}\rho\frac{\pi D^2}{4}C'_{Tn}\hat{u}_n^3. \tag{6}$$

These idealized conditions assume that the electrical power generated by the turbine is proportional to the power extracted from the flow and the control actions do not significantly affect the aerodynamic efficiency of the blades (Goit and Meyers, 2015, Appendix A). Aerodynamic losses could also be taken into account by reducing the local power coefficient $C'_P \approx \alpha C'_T$ by a constant factor $\alpha$ (Stevens and Meneveau, 2014). For example, a wind turbine operating at a thrust coefficient of $C_T = 0.75$ and $C_P = 0.45$ would use $\alpha = 0.8$. The following subsections describe the static and dynamic wake models in more detail.

## 2.1   Static wake model

The static wake model used in this work is the Jensen model with the modifications described above. In order to use this static wake model in a model-based controller for the farm power production, the model must be augmented to account for time-varying changes in the local thrust coefficient $C'_{Tn}(t)$. Including time dependency in the thrust coefficient and replacing the induction factor in Eq. (2) with the expression in Eq. (5) gives the following expression for the velocity deficit for the $n$-th

turbine

$$\delta u_n(x,t) = \frac{C'_{Tn}(t)}{4 + C'_{Tn}(t)}\frac{2U_\infty}{[1 + 2k_n(x - s_n)/D]^2}. \tag{7}$$

With this approach, thrust coefficient changes instantaneously affect the velocity deficit everywhere; i.e., the wakes implicitly have an infinitely fast advection speed. Finally, the velocity at the turbines of the $n$-th row can be found by explicitly writing out the set of upstream turbines in Eq. (3) affecting the velocity at the $n$-th turbine and using the equation for the rotor-averaged

velocity in Eq. (5)

$$\hat{u}_n(t) = \left(1 - \frac{C'_{Tn}(t)}{4 + C'_{Tn}(t)}\right)\left(U_\infty - \sqrt{\sum_{m=1}^{n-1}\delta u_m^2(s_n - s_m)}\right). \tag{8}$$

Eqs. (7) and (8) are therefore the static wake model equations $\mathbf{W}_s(\mathbf{C}'_T, \mathbf{q}_s) = \mathbf{0}$, where $\mathbf{q}_s = [\boldsymbol{\delta u}, \hat{\mathbf{u}}]$ denote the model states and boldface indicates vectors.

## 2.2   Dynamic wake model

The dynamic wake model also borrows from the classic Jensen model, but instead allows the wake velocity deficit to move downstream at a finite velocity. The resulting one-dimensional time-varying wake model, which assumes that the wake travels with the inlet velocity $U_\infty$, was previously proposed and validated against LES of wind farms at startup (Shapiro et al., 2017a). This led to the velocity deficit being governed by

$$\frac{\partial \delta u_n}{\partial t} + U_\infty\frac{\partial \delta u_n}{\partial x} = -w_n(x)\delta u_n(x,t) + f_n(x,t), \tag{9}$$

where $w_n(x)$ is the wake decay function and $f_n(x,t)$ is a forcing function used to account for the effect of the turbine on the flow field. The wake decay function

$$w_n(x) = 2 \frac{U_\infty}{d_n(x)} \frac{d}{dx} d_n(x) \tag{10}$$

is determined by assuming that the wake diameter normalized by the rotor diameter $d_n(x) = D_{wn}(x)/D$ at a fixed location $x$ is constant in time. Momentum theory shows that as the air flows through the turbine rotor, the velocity reduces to $U_\infty - 2U_\infty C'_{Tn}/(4+C'_{Tn})$ (Shapiro et al., 2017a). In order to retrieve this expected velocity reduction, the forcing function is specified as

$$f_n(x,t) = \frac{2U_\infty^2}{d_n^2(x)} \frac{C'_{Tn}(t)}{4+C'_{Tn}(t)} G(x - s_n), \tag{11}$$

where $G(x - s_n)$ is a smoothing function that integrates to unity, centered at the streamwise location of the turbine $x = s_n$. A Gaussian function with characteristic width $\Delta$

$$G(x - s_n) = \frac{1}{\Delta\sqrt{2\pi}} e^{-\frac{(x-s_n)^2}{2\Delta^2}} \tag{12}$$

maintains smoothness in the velocity deficit fields.

In the Jensen model (Katić et al., 1986), the dimensionless diameter of the wake generated by turbine row $n$ is $d_n(x) = 1 + 2k_n(x - s_n)/D$, where $k_n$ is an empirical wake expansion coefficient. We make two modifications to this equation. First, the linear expansion is assumed to begin at $x = s_n + 2\Delta$ to prevent the wake expansion from occurring within the induction zone imposed by the Gaussian forcing. The second modification addresses the fact that the equation for the standard Jensen dimensionless wake diameter is ill-posed upstream of the turbine, where it can vanish or become negative. Therefore, we use the following modified function that smoothly approximates the linear expansion in the far wake while avoiding becoming less than unity close to the turbine

$$d_n(x) = 1 + k_n \ln \left[ 1 + \exp\left( \frac{x - s_n - 2\Delta}{D/2} \right) \right]. \tag{13}$$

As in the static model, squared deficits (Katić et al., 1986) are superposed to calculate the estimated streamwise velocity $\hat{u}_n$ at the turbine

$$\hat{u}_n(t) = U_\infty - \int_0^L \left( \sum_{m=1}^N \delta u_m^2(x,t) \right)^{1/2} G(x - s_n)\,dx. \tag{14}$$

Finally, the total estimated power $\hat{P}_n$ of the $M$ turbines in row $n$ is found using Eq. (6). The dynamic wake model equations Eqs. (9)–(14) are written as $\mathbf{W}_d(\mathbf{C}'_T, \mathbf{q}_d) = \mathbf{0}$, where $\mathbf{q}_d = [\boldsymbol{\delta}\mathbf{u}, \hat{\mathbf{u}}]$ denote the model states.

## 3  Controlled wind farm designs

The model-based controllers implementing the static and dynamic wake models discussed above are designed to track the power reference signals $P_{\text{ref}}(t)$ sent by an ISO by modulating the thrust coefficients of each turbine row $C'_{Tn}(t)$. Thrust

modulation control is used as a proxy for direct actuation of blade pitch angle and generator torque. Explicit actuation of these control variables is the subject of future work. In both cases, feedback is included by measuring the row-averaged, rotor-averaged wind speed $u_n(t)$. The resulting feedback term $\epsilon_n(t)$ is fed into the controller and used to correct the predicted power output of the wake model. A block diagram of the controlled wind farm system is shown in Figure 2.

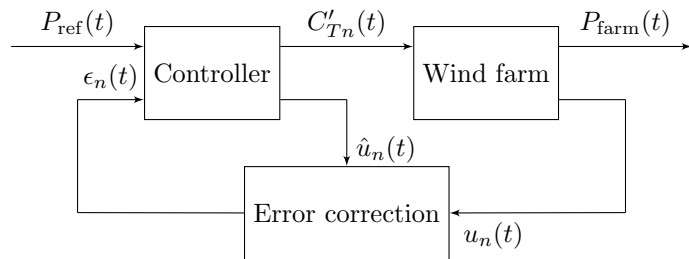

**Figure 2.** Block diagram of the controlled wind farm system for both controllers. Each controller computes a thrust coefficient command signal $C'_{Tn}(t)$ using the reference signal $P_{\text{ref}}(t)$ and an error correction term $\epsilon_n(t)$. The error correction is computed using the measured velocity $u_n(t)$ and the predicted velocity $\hat{u}_n(t)$ from the underlying wake model.

### 3.1 Controller designs

Each controller calculates the local thrust coefficient trajectories by repeatedly solving a minimization problem of the form

$$\underset{\mathbf{C}'_T, \mathbf{q}}{\text{minimize}} \quad \mathcal{J}(\mathbf{C}'_T, \mathbf{q}) + \mathcal{R}(\mathbf{C}'_T) \tag{15}$$

$$\text{subject to} \quad \mathbf{W}(\mathbf{C}'_T, \mathbf{q}) = 0, \tag{16}$$

where $\mathbf{W}(\mathbf{C}'_T, \mathbf{q}) = \mathbf{0}$ and $\mathbf{q}$ are placeholders for the static and dynamic wake model and states, which were previously indicated by the subscripts $s$ and $d$, respectively. The cost functional $\mathcal{J}$ represents the reference tracking goal, and the functional $\mathcal{R}$ contains regularizations to maintain well-behaved control trajectories. These regularizations include a penalization for fast changes in the thrust coefficients to avoid excessive oscillations in the control and a penalization for deviations away from the pre-control reference power to prevent thrust coefficients from moving outside of physical bounds.

Although both controllers solve an online optimization problem, the mechanics of the implementation are quite different. Since the equations for the static model have no dynamics, every instance in time is uncoupled. Therefore, the static-model controller can consider each point in time as a separate minimization problem, and the cost functionals can be written solely in terms of the current state. With this approach, the power tracking cost functional at time $t$ is written as

$$\mathcal{J}_s(\mathbf{C}'_T, \mathbf{q}_s) = \frac{1}{\mathcal{P}^2} \left( \sum_{n=1}^{N} \hat{P}_n(t) - P_{\text{ref}}(t) \right)^2, \tag{17}$$

and the regularization terms are

$$\mathcal{R}_s(\mathbf{C_T'}) = \eta \sum_{n=1}^{N} \left(C_{Tn}'(t) - C_{T,\text{ref}}\right)^2 + \gamma T^2 \sum_{n=1}^{N} \left(\frac{dC_{Tn}'}{dt}\right)^2. \tag{18}$$

The constants $\mathcal{P}$ and $T$ in Eq. (17)–(18) are used to make each term in the power tracking cost functional and regularization functional dimensionless and of comparable magnitude. Here we choose $\mathcal{P} = M \frac{1}{2}\rho \frac{\pi D^2}{4} U_\infty^3$ and time $T$ as the time horizon of the reference signal considered. The constants $\eta$ and $\gamma$ are the respective weights of each regularization term, which are set to $\eta = 0.005$, $\gamma = 2.083 \times 10^{-5}$ in this study.

The dynamic-model controller (Shapiro et al., 2017a), on the other hand, accounts for the time-dependent advection of turbine wakes. We therefore employ a model-based receding horizon framework, which is a predictive approach that uses the model to plan future control actions. The receding horizon method works by iteratively solving a finite-time minimization problem over a time horizon $T$. The solution is implemented for a shorter period $T_A$ before re-solving the minimization problem. More details about this procedure can be found in (Bewley et al., 2001; Goit and Meyers, 2015). With this predictive framework, the reference tracking goal is represented by the cost functional

$$\mathcal{J}_d(\mathbf{C_T'}, \mathbf{q}_d) = \frac{1}{\mathcal{P}^2 T} \int_0^T \left(\sum_{n=1}^{N} \hat{P}_n(t) - P_{\text{ref}}(t)\right)^2 dt, \tag{19}$$

and the regularization functional is defined as

$$\mathcal{R}_d(\mathbf{C_T'}) = \frac{\eta}{T} \sum_{n=1}^{N} \int_0^T \left(C_{Tn}'(t) - C_{T\text{ref}}'\right)^2 dt + \gamma T \sum_{n=1}^{N} \int_0^T \left(\frac{dC_{Tn}'}{dt}\right)^2 dt. \tag{20}$$

Consistent with the distinction between the non-predictive and predictive nature of the static-model and dynamic-model controllers, the functionals for the static wake model are not integrated forward in time. In other words, the static wake model is not a receding horizon method because the modeled system does not include dynamics.

All minimizations are solved using the modified unconstrained reduced cost functional $\tilde{\mathcal{J}}(\mathbf{C_T'}) = \mathcal{J}(\mathbf{q}, \mathbf{C_T'})$ (Bewley et al., 2001; Goit and Meyers, 2015), instead of the cost functional defined in Eq. (15). Minimizations are performed using the gradient-based nonlinear Polak-Ribière conjugate gradient method (Press, 2007) combined with the Moré-Thuente line search method (Moré and Thuente, 1994) and terminated after 100 iterations. For the static wake model, gradients are obtained using finite differencing, which can be implemented efficiently because there are only $N$ control variables per minimization. For the dynamic wake model, gradients are obtained using one backward simulation of the adjoint equations of the wake model using the formal Lagrangian method (Goit and Meyers, 2015; Borzì and Schulz, 2011). The full procedure is detailed in (Shapiro et al., 2017a). This approach was chosen because it is computationally efficient for systems with large state spaces, such as the discretized PDE system described by the dynamic wake model.

In this work, we use horizon and advancement times of $T = 40$ min and $T_A \approx 10$ s, resepctively. With these values, the optimization takes approximately 1 minute on a single processor, which is roughly six times as long as the advancement time of

$T_A = 10$ s. However, several modifications can reduce the optimization time significantly. For example, a previous implementation reduced the optimization time to a fraction of the advancement time by employing a quasi-Newton minimization method, reducing the horizon and advancement times, and limiting the number of minimization iterations (Shapiro et al., 2017b). As a result, this approach is feasible for real time control.

## 3.2 Measurement feedback

As shown in Figure 2, controllers employ closed-loop feedback for velocity measurements at each turbine to correct modeling errors and assumptions. The row-averaged power and row- and rotor-averaged wind velocities are defined as

$$P_n = \sum_{m=1}^{M} P_{nm}, \quad \text{and} \quad u_n = \frac{1}{M} \left( \sum_{m=1}^{M} u_{nm}^3 \right)^{1/3}, \tag{21}$$

where $u_{nm}$ is the velocity measured at the turbine in the $n$-th row and $m$-th column of the wind farm. The definition of the row-average velocity at the turbine disk is necessary to ensure that $P_n = M \frac{1}{2} \rho \frac{\pi D^2}{4} C'_{Tn} u_n^3$. These measurements are used to calculate an error term $\epsilon_n$ and provide feedback by replacing Eq. (6) with

$$\hat{P}_n = M \frac{1}{2} \rho \frac{\pi D^2}{4} C'_{Tn} (\hat{u}_n + \epsilon_n)^3. \tag{22}$$

For the static wake model, the error term for turbine row $n$ at time step $k$ is calculated using the difference between the measured and estimated velocity from the previous time step $k-1$

$$\epsilon_n^k = u_n^{k-1} - \hat{u}_n^{k-1}. \tag{23}$$

For the dynamic wake model, the error correction at the receding horizon iteration starting at time $t_c$ is

$$\epsilon_n(t) = (u_n(t_c) - \hat{u}_n(t_c)) e^{-(t-t_c)/\tau}. \tag{24}$$

The exponential decay accounts for the reduced future accuracy of the error term in the receding horizon prediction and is set to $\tau = 120$ s in this study.

## 4 Virtual wind farm test system

A LES model of a wind farm with wind turbines represented using actuator disk models is used to test the two control approaches. The wind farm is composed of $N = 7$ rows of $M = 12$ aligned columns of turbines. Each turbine has a 100 m rotor diameter $D$ and a 100 m hub height. The spacing between turbines is $7D$ in the streamwise direction and $5D$ in the spanwise direction. Prior to the initiation of the control actions, all of the turbines are operated at a constant reference local thrust coefficient of $C'_{T,\text{ref}} = 1.33$, which is assumed to be representative of wind turbines operating in region 2 (Calaf et al., 2010; Stevens et al., 2014a).

These simulations are performed using JHU's LES code LESGO (Calaf et al., 2010; Stevens et al., 2014b; VerHulst and Meneveau, 2015), which uses pseudo-spectral discretization in the horizontal directions with periodic boundary conditions. The

code also employs second-order Adams-Bashforth time integration, second-order finite differencing in the vertical direction, and the dynamic scale-dependent Lagrangian Smagorinsky subgrid stress model (Bou-Zeid et al., 2005). Inlet conditions for the wind farm are generated using the concurrent-precursor method (Stevens et al., 2014b). The force exerted by and the power extraction of the $m$-th turbine of the $n$-th row are both a function of the filtered rotor-averaged velocity $u_{nm}$ (Calaf et al., 2010)

and the thrust coefficient $C'_{Tnm}$. The force is modeled as a drag force $F_{nm} = -\frac{1}{2}\rho\frac{\pi D^2}{4}C'_{Tnm}u_{nm}^2$, and the power extraction is $P_{nm} = -F_{nm}u_{nm}$. An instantaneous color contour plot of the streamwise velocity field from one of these simulations is shown in Figure 3.

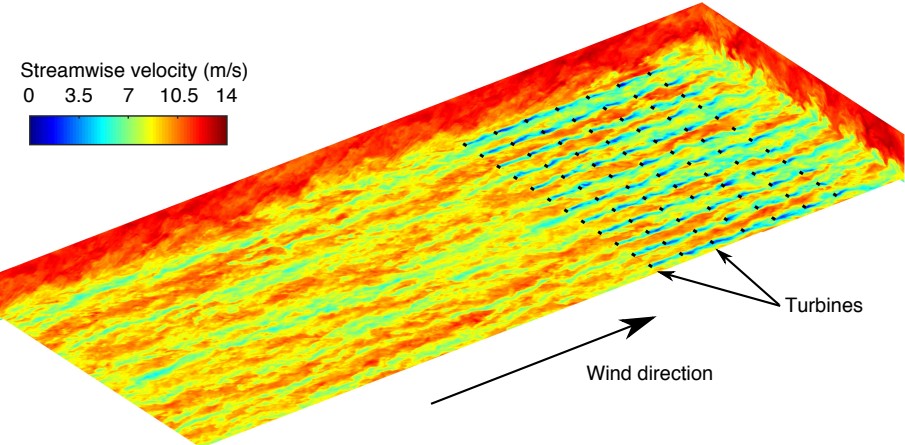

**Figure 3.** Instantaneous streamwise velocity contours for a large eddy simulation with actuator disk turbine models, which are indicated by black dashes. Each turbine has a rotor diameter of $D = 100$ m and hub height of 100 m. The mean and maximum inlet velocities are approximately 9.5 m/s and 12 m/s, respectively. The inlet conditions are generated using a concurrent precursor simulation (Stevens et al., 2014b) shown at the beginning of the plotted domain.

## 5   Test cases

The performance of the controlled wind farms is evaluated using PJM's published RegA and RegD test signals as well as
historical RegA and RegD signals from three hours in 2015 (PJM, 2012, 2015, 2016). For each regulation signal, we use three initial conditions for the wind farm and two levels of power setpoint reduction, which we also refer to as derates. For each controller test, the reference signal is defined as $P_{\text{ref}}(t) = [1 - \alpha + 0.08r(t)]P_{\text{base}}$, where $P_{\text{base}}$ is the 5-minute average power prior to initiation of the control, $\alpha$ is the derate amount, and $r(t) \in [-1, 1]$ is the regulation signal from the ISO. As a result, the reference signal varies by $\pm 8\%$ of the baseline power $P_{\text{base}}$.

The combination of test signals and initial conditions lead to 48 unique test cases, each of which is given a unique identifier that is a combination of identifiers for each of the variable types shown in Table 1. "Signal" refers to the regulation signal type (RegA or RegD), "Derate" refers the to derate amount (4 or 6%), "Initial condition" refers to the initial condition of the

controlled plant simulation, and "Period" refers to the regulation signal period, which is either the PJM test signals or one of the selected hours in 2015. For example, the test case "RegA.D4.IC1.TS" refers to the case with the RegA test signal, 4% derate, and the first initial condition.

**Table 1.** Test case identifiers describing the signal type, derate amount, initial condition of the wind farm, and regulation signal period. For example, the test case "RegD.D6.IC1.H2' refers to the case with the second RegD historical signal, 6% derate, and the first initial condition.

| Identifier | Type | Description |
|---|---|---|
| RegA | Signal | Traditional RegA regulation signal |
| RegD | Signal | Fast-responding RegD regulation signal |
| D4 | Derate | Power setpoint is reduced by 4% of $P_{\text{base}}$ |
| D6 | Derate | Power setpoint is reduced by 6% of $P_{\text{base}}$ |
| IC1 | Initial condition | Initial condition 1 |
| IC2 | Initial condition | Initial condition 2 |
| IC3 | Initial condition | Initial condition 3 |
| TS | Period | PJM test signals |
| H1 | Period | PJM historical hour 1 |
| H2 | Period | PJM historical hour 2 |
| H3 | Period | PJM historical hour 3 |

## 5.1 Historical PJM regulation signals

The number of historical hours used to test the controlled wind farm is constrained by the computational cost of running the model wind farm LES. As a result, it is impractical to select enough hours to sample the entire range of possible regulation signals provided by PJM. To prevent systematic bias, the three hours were selected without considering the characteristics of the regulation signals during those periods.

In order to evaluate whether the slected signals are representative cases, we compare them to the range of all possible regulation signals provided by PJM in 2015 using three statistics

$$\mathcal{S}_1 = \frac{1}{T}\int_0^T r(t)\,dt \qquad \mathcal{S}_2 = \frac{1}{T}\int_0^T r^2(t)\,dt - \mathcal{S}_1^2 \qquad \mathcal{S}_3 = \frac{1}{T}\int_0^T \left(\frac{dr}{dt}\right)^2 dt, \tag{25}$$

where $r(t)$ is the regulation signal, $T = 60$ min, $\mathcal{S}_1$ is the mean of $r(t)$, $\mathcal{S}_2$ is the variance of $r(t)$, and $\mathcal{S}_3$ is the variance of $\frac{dr}{dt}$. For each of these statistical measures, the probability density function (PDF) is calculated using all possible hourly signals provided by PJM in 2015 and is shown in Figure 4. These PDFs demonstrate the differences between the RegA and RegD signals. The RegA signals have a larger mean and variance than the RegD signals, but the variance of $\frac{dr}{dt}$ is smaller. The values of these statistics for the three selected hours is compared to the PDFs over the entire year in Figure 4. These figures show that

the selected historical signals represent a reasonable cross section of the possible PJM regulation signals. The only exception, the high percentile ranking in $\mathcal{S}_1$ of the RegA signals, represents a more difficult test for the controlled wind farm because more energy is requested than the average.

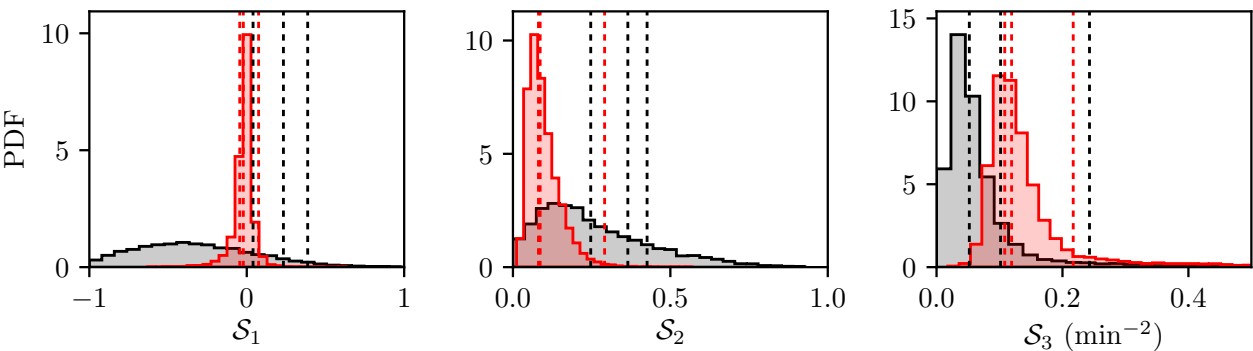

**Figure 4.** Probability density functions of $\mathcal{S}_1$–$\mathcal{S}_3$ defined in Eq. (25) for RegA (black) and RegD (red) during 2015. The three selected historical hours are shown in the PDFs as vertical dashed lines.

## 5.2 Wind farm initial conditions

We set the initial conditions of the controlled wind farm simulations to correspond to uncontrolled simulations with a local thrust coefficient of $C'_{T,\mathrm{ref}} = 1.33$, as previously discussed. The inflow characteristics for the three initial conditions of interest are provided in Table 2. The inflow velocities of the initial conditions have a mean $\overline{u} \approx 9.5$ m/s and a standard deviation $\sigma_u \approx 1$ m/s as measured at the first row of turbines during the $T_0 = 5$ min prior to initiation of the control

$$\overline{u} = \frac{4 + C'_{T,\mathrm{ref}}}{4} \frac{1}{T_0 M} \int\limits_{-T_0}^{0} \sum_{m=1}^{M} u_{1m}(t)\,dt \qquad \sigma_u = \frac{4 + C'_{T,\mathrm{ref}}}{4} \left[ \frac{1}{T_0 M} \int\limits_{-T_0}^{0} \sum_{m=1}^{M} (u_{1m}(t) - \overline{u})^2\,dt \right]^{1/2}. \tag{26}$$

The turbulence intensity as measured at the center of each of the turbine rotors is approximately 13%, which corresponds to low to medium IEC turbulence levels (IEC, 2005). The simulations assume region 2 operation (Johnson et al., 2006) with idealized aerodynamic characteristics of $C'_P = C'_T$. In order to avoid any significant interaction with the rated regime, we presume wind turbines with a rated wind speed of at least 12m/s, which corresponds to the 99th percentile of the LES velocity field. Wind turbines with a diameter of $D = 100$ m and a power coefficient of $C_P = 0.5625$, which corresponds to $C'_P = C'_T$, therefore

have a rated power of approximately 4.5 MW and an average total farm power of approximately 100 MW. Under non-ideal aerodynamic conditions ($C'_P = 0.8 C'_T$ cf. Section 2; Stevens et al., 2014a), a power coefficient of $C_P = 0.45$ would yield a rated wind turbine power of 3.6 MW.

The required parameters of the static and dynamic wake models, inlet velocity $U_\infty$ and wake expansion coefficients $k_n$, are also calculated for each initial condition using measurements from the $T_0 = 5$ min prior to initialization of the control. The inlet velocity is set using the relationship $U_\infty = \frac{1}{4}(4 + C'_{T,\text{ref}})T^{-1}\int_{-T_0}^{0} u_1(t)\,dt$, and the wake expansion coefficients are found using a least squares fit between the measured power and the power predicted by the static model. Note that the inlet velocity for the model is defined using the average power and therefore the average inflow velocity is not equal to the inlet velocity for the models $\overline{u} \neq U_\infty$. The resulting parameters are also shown in Table 2.

**Table 2.** Characteristics of wind farm initial conditions, including mean inlet velocity $\overline{u}$, standard deviation of inlet velocity $\sigma_u$, and turbulence intensity TI. The corresponding wake model inlet velocity $U_\infty$ and wake expansion coefficients $k_n$ are also shown.

| Initial condition | $\overline{u}$ (m/s) | $\sigma_u$ (m/s) | TI (%) | $U_\infty$ (m/s) | $k_1$ | $k_2$ | $k_3$ | $k_4$ | $k_5$ | $k_6$ | $k_7$ |
|---|---|---|---|---|---|---|---|---|---|---|---|
| 1 | 9.53 | 1.12 | 13.6 | 9.65 | 0.028 | 0.049 | 0.041 | 0.047 | 0.053 | 0.054 | 0.054 |
| 2 | 9.22 | 0.97 | 13.3 | 9.32 | 0.026 | 0.046 | 0.043 | 0.047 | 0.054 | 0.052 | 0.052 |
| 3 | 9.56 | 0.93 | 12.5 | 9.64 | 0.026 | 0.040 | 0.040 | 0.037 | 0.044 | 0.041 | 0.041 |

## 6 Comparison of control methods

The power tracking performance and control trajectories of the controlled wind farm, represented by the LES described in Section 4, are shown in Figures 5 and 6. The left and right panels of these figures show the performance of the static and dynamic-model controllers, respectively. Figure 5 shows the response of the controlled farms to the RegA test signals, and Figure 6 shows the response of the controllers to the RegD test signals. The dynamic-model control demonstrates good overall tracking performance, although it has some trouble tracking the reference signal during the last 5–10 minutes of the RegA.D4.IC1.TS and RegA.D4.IC3.TS cases. On the other hand, the static-model control demonstrates poor overall tracking performance, although it is able to track the signal for certain down regulation events, e.g. around minute 20 in all cases in Figure 5.

The static-model control method appears to switch between two distinct operating points, depending on the characteristics of the regulation signal. Down-regulation trajectories are often successfully tracked by increasing the thrust coefficient of the first row of turbines to values above $C'_T = 2$. This change in operating conditions increases the magnitude of the velocity deficits throughout the farm, thereby reducing the overall wind speed and total power production. When there is a period of up-regulation approaching or the wind farm is slightly underproducing, the controller quickly reduces the upstream thrust coefficients and moves to the Betz-optimal thrust coefficient $C'_T = 2$ (Goit and Meyers, 2015) for the last row. This operating point is likely the optimal power point for the Jensen model with constant wake expansion coefficients.

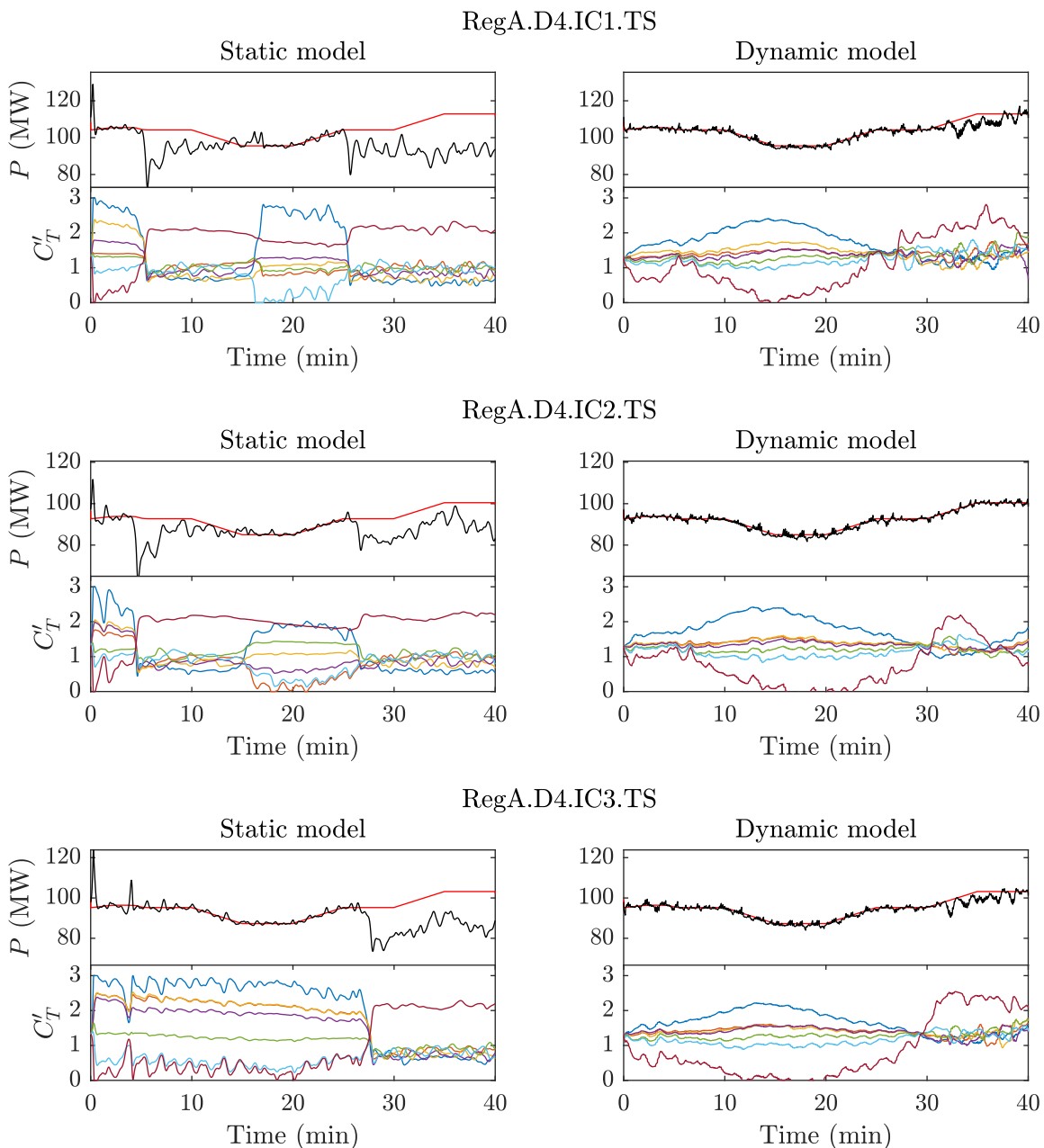

**Figure 5.** Comparison of static-model (left) and dynamic-model (right) control methods for RegA test signals with 4% derates. All three initial conditions 1–3 are shown from top to bottom. The top panel in each row shows the controlled LES wind farm model power production (——) compared to the reference signal (——). The bottom panel in each row shows the local thrust coefficients calculated by control methods by row: row 1 (——), row 2 (——), row 3 (——), row 4 (——), row 5 (——), row 6 (——), row 7 (——).

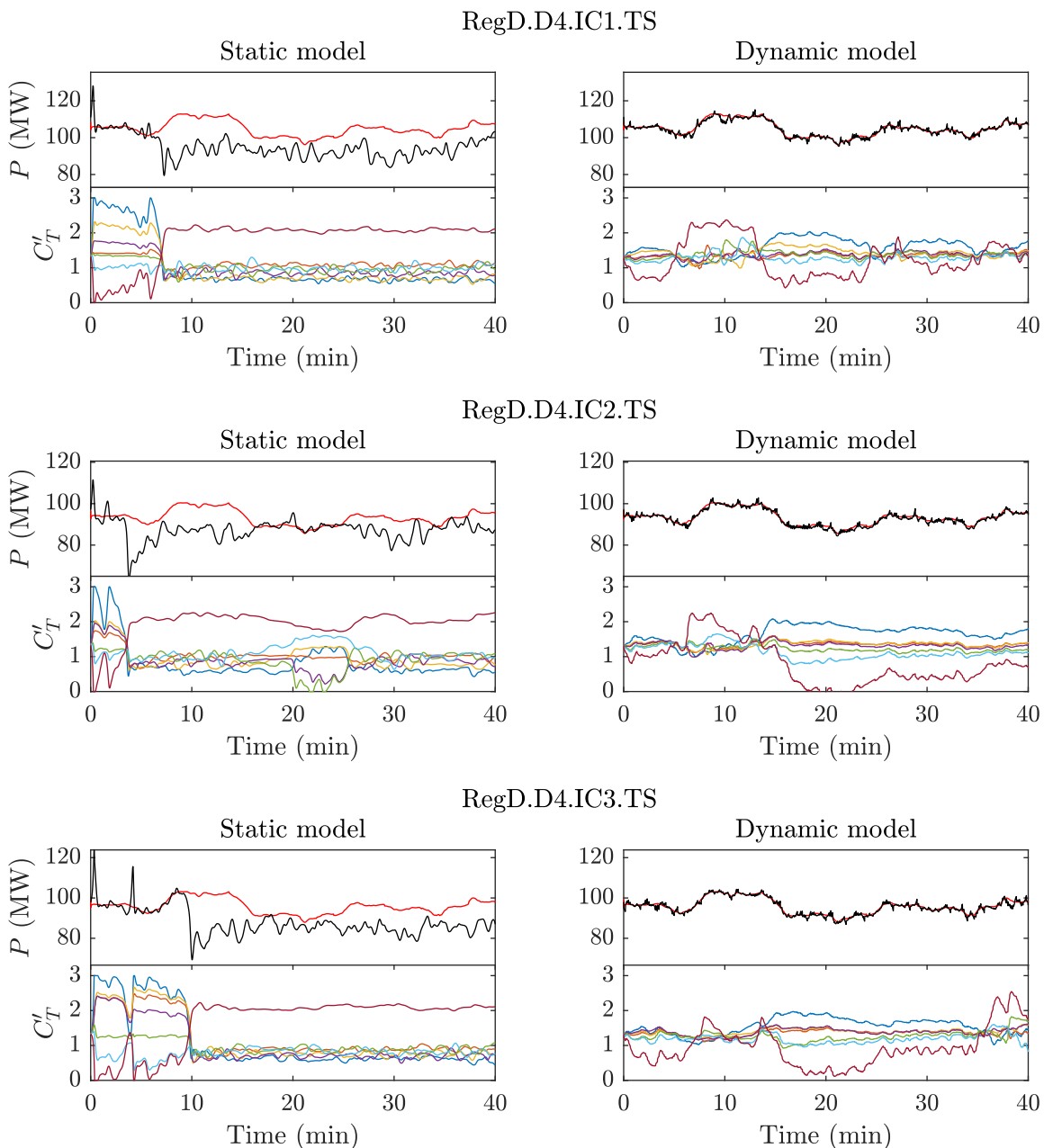

**Figure 6.** Comparison of static-model (left) and dynamic-model (right) control methods for RegD test signals with 4% derates. All three initial conditions 1–3 are shown from top to bottom. The top panel in each row shows the controlled LES wind farm model power production (——) compared to the reference signal (——). The bottom panel in each row shows the local thrust coefficients calculated by control methods by row: row 1 (——), row 2 (——), row 3 (——), row 4 (——), row 5 (——), row 6 (——), row 7 (——).

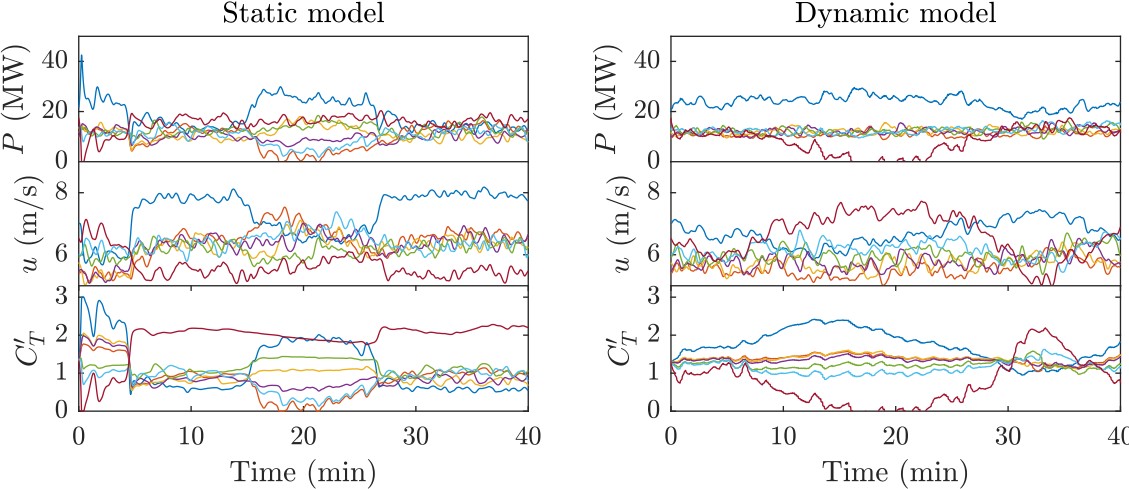

**Figure 7.** Comparison of static-model (left) and dynamic-model (right) control methods for RegA.D4.IC2.TS simulation case. Each panel shows the controlled LES wind farm model power production, rotor-averaged velocity, and thrust coefficients by row: row 1 (——), row 2 (——), row 3 (——), row 4 (——), row 5 (——), row 6 (——), row 7 (——).

The performance of the static-model control provides an interesting demonstration of the importance of including time dependency in the wake model used in this type of control scheme. In an attempt to track the changing reference signal, the controller switches quickly between the two operating points discussed above. The static Jensen model erroneously models these transitions between operating points as an instantaneous change of the wake velocity deficit throughout the farm. In
reality, the air around the turbine will slowly respond to a sudden change in the thrust coefficient and the reduced wake deficit must travel through the farm before the effects of changing upstream thrust coefficients on downstream power production and wind speeds are realized. Detailed trajectories of the power and rotor-averaged velocity of each row in Figure 7 show that the LES wind farm does not respond instantaneously to the change in operating point. Instead, power production slowly increases between minutes 5 and 15.
As a result of these modeling errors, the static-model controller produces large transient variations in power production when moving between operating points. When moving to the up-regulation operating point identified by the controller, the power production of the farm plunges. In some cases, the total power production slowly recovers to the desired setpoint. Furthermore, all of the static-model control cases in Figures 5 and 6 demonstrate significant overshoot in the power production during the first 30 seconds of the simulations as the thrust coefficients quickly move away from the pre-control level. The frequency of
the changes in the control actions, which is determined by the advancement time of approximately 10 s, is faster than these dynamics of wake advection and is therefore unlikely to explain these effects. Instead, neglecting the wake advection time in the Jensen model best explains the poor performance of the static model controller.

The dynamic-model control uses strategies similar to those of the static-model controller, including increasing the thrust coefficient during down-regulation periods and moving toward a Jensen model optimal power point for up-regulation periods. However, by including the time-dependent effects of wake advection, the controller avoids large transient changes when changing between states. The underlying dynamic model can correctly predict the time-varying effect of changing upstream thrust coefficients on downstream power production. In the next section we further study the performance of this dynamic-model control approach.

## 7   Performance evaluation of dynamic-model control

The time evolution of the total LES wind farm power is compared to the reference signals for initial condition 3 and a 4% derate in Figure 8, which shows all regulation signals (RegA or RegD) and regulation period combinations. The controlled wind farm power production is also compared to the uncontrolled case, where the wind farm is kept at the constant pre-control thrust coefficient. These results demonstrate the good overall tracking performance of the controlled wind farm, except for a few specific periods of under-performance. Furthermore, the results demonstrate that the dynamic-model based receding horizon control method is also able to reduce the natural turbulent fluctuations in the wind farm power production. Indeed, the root-mean-square (RMS) of the controlled wind farm power production about the reference signal is 1.06 MW, which is almost a quarter of the 3.93 MW RMS of uncontrolled power production about the baseline power.

Quantitative measures of the performance of each regulation signal type (RegA or RegD) for derate values of 4% and 6% are shown in terms of PJM's performance scores in Figure 9. In order to participate in PJM's regulation market, power plants must pass the Regulation Qualification Test for the particular regulation signal being supplied. This test is carried out over a 40-minute period, and the tracking capability is quantified using a composite performance score, which is the weighted sum of accuracy, delay, and precision scores (PJM, 2012, 2015). The accuracy score measures the ability of the signal to respond to a change in the ISO regulation signal. The delay score measures the delay in the plant's response to the regulation signal. The precision score measures the difference between the requested power and the plant's power output. A minimum score of 75% is needed to qualify to participate in each of the two regulation services. Once qualified for a particular service, a plant is continuously evaluated; if its average score over the last 100 hours drops below 40%, then the plant is disqualified from providing the service and must retake the initial performance test to requalify.

The controlled wind farm performs better for the RegD signals, meeting the composite score threshold for qualification of 75% in all cases. The performance of the controlled farm in tracking the RegA signals is also satisfactory for PJM participation, but the controlled farm would not have qualified in all tests. These lower composite scores may be explained by the large values in $\mathcal{S}_1$, which represent the total energy requested in the signals, compared to other PJM signals. However, in cases where the controlled wind farm had poor performance for the RegA signal with 4% derate, increasing the derate to 6% markedly improved the overall performance.

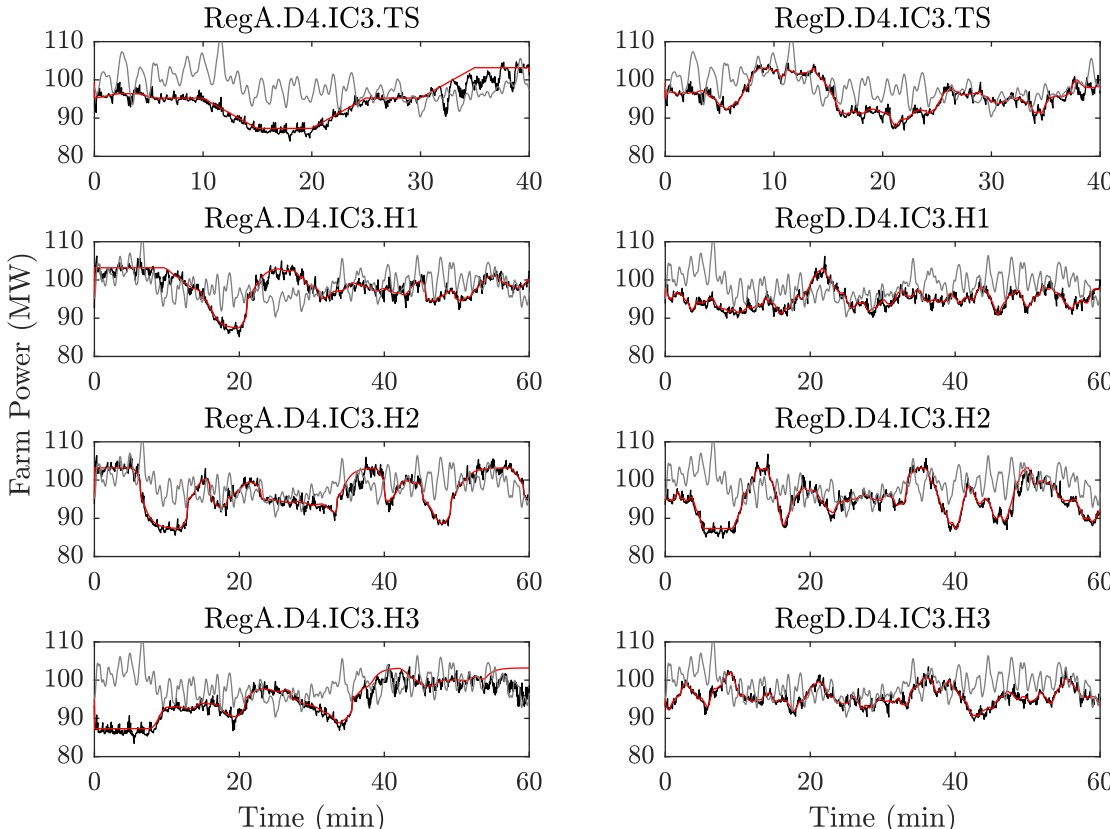

**Figure 8.** Power tracking performance of dynamic-model controlled wind farms comparing simulated farm power from a controlled LES wind farm model (——), an uncontrolled LES wind farm model (——), and power reference signals (——) for 4% derates and initial condition 3.

The results shown in Figures 5–9 provide important insights into the possible strengths and limitations of the proposed approach to wind farm control for frequency regulation. These results suggest that wind farms may be well suited to act as a quickly responding resource for grid regulation services. For example, the consistent passing of the composite performance score for the RegD signals indicates that dynamic-model controlled wind farms are able to provide this service reliably.

The power tracking results in Figure 8 demonstrate that the controller is able to track the up-regulation portions of the RegA signals at the beginning of the control period, such as during the first 5–10 minutes of the first two historical signals. In several cases the controlled LES wind farm is able to produce more power than the uncontrolled case, such as after minute 20 of the "RegA.D4.IC3.H1" and "RegD.D4.IC3.H1" simulations. However, when up-regulation is requested for prolonged periods or towards the end of the control interval, such as the last 10 minutes of the "RegA.D4.IC3.TS" and "RegA.D4.IC3.H3" cases, the controller does not perform as well. A possible explanation is that the available energy in the wind is slowly changing as the atmospheric boundary layer evolves, as demonstrated by the declining power production of the uncontrolled simulations

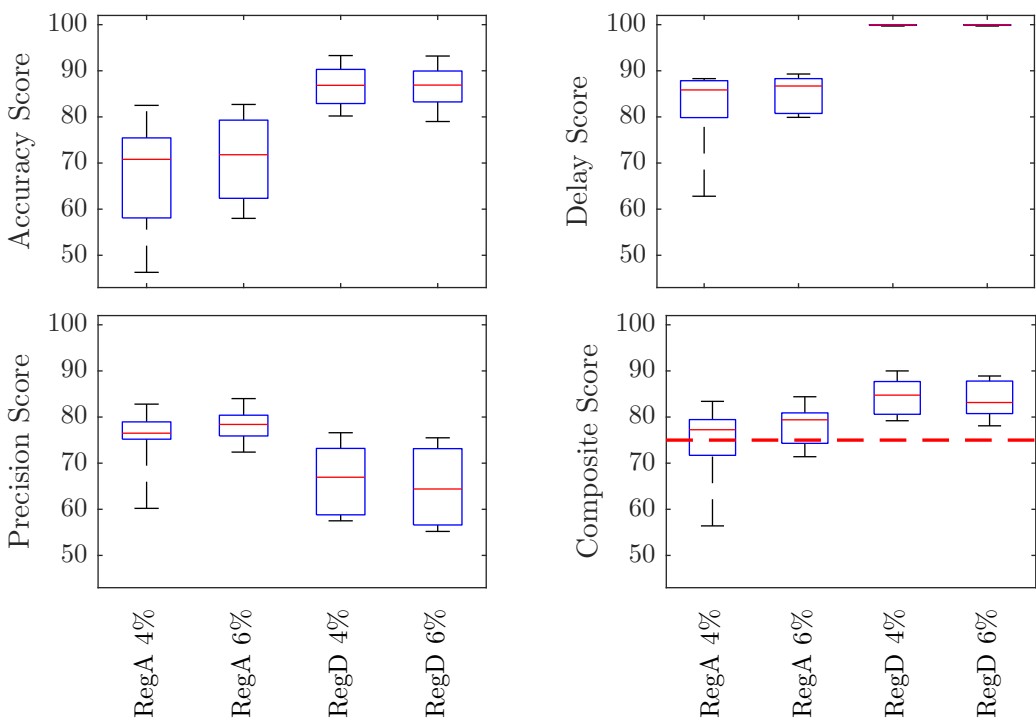

**Figure 9.** Boxplots of PJM performance scores for dynamic-model controlled wind farm for all regulation signal types (RegA or RegD) with derate values of 4% and 6%. The qualification threshold of 75% for the composite score (— — —) is shown in the lower right panel. The average controlled system performance exceeds the 75% for both signal types, but only the RegD cases pass all of the time.

during these time periods. Since estimates of available energy are readily available over short time horizons, more frequent market clearing may allow wind farms to more effectively provide regulation. Ultimately, future work is needed to determine whether this is a fundamental limitation of the wind farm dynamics or the control strategy.

## 8   Conclusions

5   In this study we further characterize the performance of wind farms providing secondary frequency regulation using the dynamic-model control framework proposed in (Shapiro et al., 2017a). This model-based receding horizon approach relies on a simple one-dimensional time-varying wake model to provide thrust coefficient trajectories for individual turbines within a wind farm. As in previous work, the control approach is tested using a "virtual wind farm" represented by LES of an 84-turbine wind farm with turbines modeled as actuator disks.

10   First, we evaluate the relative importance of including the dynamics of wake advection in the control scheme by comparing the performance of the dynamic-model controller to a comparable static-model controller. Tests using regulation signals from

PJM indicate that the dynamic-model control demonstrates good overall tracking performance, whereas static-model control failed to match the reference signal for all simulated cases. These results indicate that the complexity of including the dynamics of wake advection is indeed required in model-based coordinated wind farm controls.

The tracking performance of the dynamic-model control method is then further quantified using PJM's performance metrics. Tests for both regulation signal types, RegA and RegD, exceed the PJM threshold for regulation participation on average, but the only the RegD signal exceeds the threshold in all cases. These results indicate that this model-based receding horizon controller design could allow wind farms to meet industry design standards and allow wind farms to fully participate in regulation markets, particularly in fast-acting regulation markets.

The potential for reducing the derate required to participate in these regulation markets was also explored. Participating in frequency regulation markets currently requires a trade-off between revenue losses in the bulk power market and revenues generated in the frequency market. Previous approaches (Aho et al., 2013; Jeong et al., 2014) required power setpoint reductions of an amount equal to the regulation amount, which directly reduces bulk power revenue by this amount. For this study we took a more aggressive approach by reducing the power setpoint by only 75%, and even only 50%, of the maximum regulation provided. For both of these derates, the controller is able to track fast-acting RegD signals. The potential for reducing the required derate has important economic implications for wind farms participating in both energy and regulation markets, a situation that will become increasingly common as more ISOs require wind farms to contribute to this grid service.

Although the dynamic-model controller design showed promising results, more work is needed to push this approach towards the implementation phase. We used the local thrust coefficient as a surrogate for real turbine control variables, such as generator torque and blade pitch angle. Improvements to our representation of these variables through actuator line methods and the inclusion of drivetrain dynamics in the control method are needed. Including rotational inertia may allow for further reductions in the amount of derate because rotational kinetic energy can compensate for short term power shortages (De Rijcke et al., 2015). Furthermore, we assumed that the ISO provided the regulation signal at the beginning of the control period; however, PJM provides this reference at a 2 second scan rate. This shortcoming could be addressed by adding estimated reference trajectories to the control design. Finally, a systematic study of the relative advantages of all of the emerging control designs for wind farms to provide secondary frequency regulation (e.g. van Wingerden et al., 2017) is needed to identify which strategies are appropriate under various market, geographic, and technical constraints.

## 9    Data availability

Data from the simulations, including the disk-averaged velocity, local thrust coefficient, and disk-center velocity, are provided in the data repository (citation to be provided).

*Author contributions.* C. Shapiro designed the study, performed the simulations, analyzed results, and wrote the first draft of the paper. J. Meyers, C. Meneveau, and D. Gayme helped design the study, analyzed results, and contributed to writing the paper.

*Competing interests.* J. Meyers is a member of the editorial board of the journal.

*Acknowledgements.* C. Shapiro, C. Meneveau, and D. Gayme are supported by the National Science Foundation (grant nos: ECCS-1230788, CMMI 1635430, and OISE-1243482, the WINDINSPIRE project). J. Meyers is supported by the European Research Council (ActiveWind-Farms, grant no: 306471). This research project was conducted using computational resources at the Maryland Advanced Research Computing Center (MARCC).

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
