# Peer review of "Wind farms providing secondary frequency regulation: Evaluating the performance of model-based receding horizon control\"

_Wind Energy Science, 2016_

## Referee Comment (RC1) · Anonymous Referee #1 · 13 Jun 2017

General comments:

This paper presents results of model predictive control of wind farms to provide secondary frequency regulation balancing services for the power grid. A time-varying onedimensional wake model is presented to model wake advection and wake interactions while trying to enable real-time implementation.

Simulations show that the time-varying wake model leads to much better results than the static wake model that is presented and evaluated as a comparison. To fully put the simulation example in context, it would be useful to know the rated power of each of the 84 wind turbines in the example wind farm. Further, what is the rated wind speed

for these turbines, what is the mean wind speed of the incoming flow onto the wind farm, and what is the distribution of the incoming wind speeds onto the wind farm in the simulations? What is the turbulence intensity? Or perhaps it is more useful to characterize the turbulence in terms of IEC turbulence characteristics.

In discussing the results shown on the left side of Figure 5, the authors describe the change in behavior at approximately 5 minutes, though they don't explain why the change in behavior occurs. Can the authors determine a reason for this sudden shift in behavior? It would be useful for readers if the authors also explain the other changes in behavior that are evident, such as around 15 minutes in the upper left 2 plots and around 25 minutes in all of the left plots.

Similarly for the right hand plots in Figure 5. The changes in behavior are slower, but there still appear to be qualitative changes in behavior. For instance, in the lower right plot, the behavior before about 29.5 minutes is different from after that time. Can the authors analyze their data further to explain why the change in behavior occurs? And of course similarly for the other right hand plots of Figure 5. And for the plots in Figure 6 as well. By understanding the reasons for the shifts in behavior, both the authors and readers will be able to gain better insight into the properties of the receding horizon control technique used in this paper.

Specific comments:

1. The second sentence of Section 3.2 does not make sense (Page 9, line 9). It reads: "The row-averaged power and row-averaged, rotor-averaged are defined from velocities  $u_{nm}$  measured at every turbine in the wind farm ... "

Would the following be more accurate? "The row-averaged power and row-averaged, rotor-averaged downstream wind velocities are defined from velocities  $u_{n}$  measured at every turbine in the wind farm ... "

2. When discussing Figure 7, the authors repeatedly specifically indicate that these
results are based on an LES simulation, while such comments are never mentioned when discussing the results in Figures 5 and 6. Presumably the results in Figures 5 and 6 are also based on LES simulations?

Technical corrections:

1. In equation (7), should the last term in the denominator be divided by D rather than multiplied by D? That is, should the last factor in the denominator be [ 1 + 2 k\_n (x - s\_n) / D ] ^2 ?

2. In the summation in equation (8), should it be delta  $u^2_m$ ? That is, should the subscript on delta u be m rather than n?

3. In the caption for Figure 3, the actuator disk turbine models look to me to be indicated by black "dashes" rather than "lines".

- 4. Page 12, line 17: "form" should be "from"
- 5. Page 13, line 1: "RegA.D4.IC4.TS" should be "RegA.D4.IC3.TS"

6. Page 13, line 29: "compared to other PJM signal" should be "compared to other PJM signals"

7. Page 16 line 6: What are "2/14" and "8/17"? These have no meaning to me.

8. References: Please list out each of the authors and do not use "et al." in any of the author lists.

---

## Referee Comment (RC2) · Anonymous Referee #2 · 25 Jun 2017

The paper presents interesting results regarding the ability of wind farms to provide secondary frequency regulation while minimizing the amount of energy not produced. Some points that would improve, in my opinion, the paper:

1. The presented wind farm control approach is likely to be computationally too expensive for the use in real wind farms. It would be useful to also discuss how the control approach could be applied to real wind farms. 2. The introduction could be shortened by moving some of the content to a methodology chapter. 3. Switching chapter 2 and 3 would improve the flow of the paper. 4. The use of thrust coefficient as input to a wind turbine controller is not realistic. 5. Please provide more details regarding the

rated power of the wind turbines, their rated wind speed and the mean wind speed considered in the simulations. 6. Please mention the frequency of the control with regards to the discussion on p. 13 line 13ff 7. Instead of showing the performance of the static model-based controller for all cases it would be useful to focus on a single cases and include figures on rotor effective wind speed at a column of turbines. 8. The impact of the paper would be improved by a comparison of the performance of the controllers to a standard PI(D) control approach. 9. In chapter 7 please use a quantitative assessment of the controller instead of qualitative statements. It is mentioned that the controller reduces turbulence driven power fluctuations. It would be necessary to justify this state by comparing the controller against a PI(D) control approach. 10. Please also include the total available wind farm power in the figures. This would also facilitate the discussion on page 16 line 8ff.

---

## Author Comment (AC2) · 10 Jul 2017

We thank the referee for their careful reading of the manuscript. We have included detailed responses to each of the referee remarks/questions below, where the referee comments are in bold face.

The paper presents interesting results regarding the ability of wind farms to provide secondary frequency regulation while minimizing the amount of energy not produced. Some points that would improve, in my opinion, the paper:

1. The presented wind farm control approach is likely to be computationally too

**expensive for the use in real wind farms. It would be useful to also discuss how the control approach could be applied to real wind farms.**

In the implementation used in this paper, the optimization step takes  $\sim$ 60 seconds on a single processor. While this is 6 times larger than the advancement time of 10 s, several refinements can bring the optimization time to a fraction of the advancement time (allowing real time control). We discuss these refinements in a recent American Control Conference paper, and we will include this discussion at the end of Section 3.1.

**2. The introduction could be shortened by moving some of the content to a methodology chapter.**

Thank you for this suggestion. We will either move page 3, lines 5–21 to section 5 or add an additional section between sections 4 and 5 discussing the PJM signals (this would include page 3, lines 5–21, and Figure 4 and the related analysis in section 5)

**3. Switching chapter 2 and 3 would improve the flow of the paper.**

We think section 3 requires information from section 2 to be fully coherent and will therefore keep the current organization.

**4. The use of thrust coefficient as input to a wind turbine controller is not realistic.**

We agree using the thrust coefficient as control input is a significant simplification. However, we discuss the future work needed to move toward more realistic controls in the last paragraph of the conclusion (page 18, line 15). To help with this discussion, we will add a sentence about the relationship between the thrust coefficient and blade pitch/generator torque on page 7, line 8.

**5. Please provide more details regarding the rated power of the wind turbines, their rated wind speed and the mean wind speed considered in the simulations.**

The actuator disk model used to represent the wind turbines in LES assumes an idealized wind turbine operating in region 2 (always operating below the rated wind speed and power). Based on the maximum observed wind speed in the simulations, we will report the effective rated wind speed and power corresponding to this assumption. These will be provided along with the mean effective wind speeds in Table 2. We will also add rotor diameter, hub height, and representative mean and maximum wind speeds to the caption of Figure 3.

**6. Please mention the frequency of the control with regards to the discussion on p. 13 line 13ff**

Control actions are applied every 10 seconds (the receding horizon advancement time). To clarify, we will add the advancement and horizon times to page 12, line 15 and mention on page 13, line 13 that the time length discussed is several times the advancement time.

**7. Instead of showing the performance of the static model-based controller for all cases it would be useful to focus on a single cases and include figures on rotor effective wind speed at a column of turbines.**

We believe it is useful to show all of the comparison cases to highlight consistent trends in the results. Since the inflow is different between the different cases, we can get a better sense of the overall performance by looking at a variety of cases. We propose adding an additional figure to show more details of row power, rotor effective wind speed, and thrust coefficient for a single simulation case.

**8. The impact of the paper would be improved by a comparison of the performance of the controllers to a standard PI(D) control approach.**

We agree that a comparison to other control designs, particularly those of Aho et al. (2013) and van Wingerden et al. (In press), would be of interest. However, a standard for farm level frequency regulation has yet to be reached and PI(D) control techniques for frequency regulation are still being developed (see van Wingerden et al.). We think

СЗ

this comparison would be well covered by a collaborative effort among the community, but it is outside the scope of the present paper. In this paper, we compare the performance of the dynamic and static model-based control designs and demonstrate that controllers based on static wake models have difficulties providing frequency regulation similar to approaches that do not consider wake effects at all (see Fleming et al., 2016).

**9. In chapter 7 please use a quantitative assessment of the controller instead of qualitative statements. It is mentioned that the controller reduces turbulence driven power fluctuations. It would be necessary to justify this state by comparing the controller against a PI(D) control approach.**

In Figure 8 of section 7 we provide a quantitative analysis of the controller using PJM's performance scores. To justify the statement about reduced turbulence driven power fluctuations, we will compare on page 13, line 24 the variance of the pre-control power about the baseline power  $P_{base}$  to the wind farm power during the controlled period about the reference signal. A comparison to other control approaches, such as PI(D), is outside the scope of this paper.

**10. Please also include the total available wind farm power in the figures. This would also facilitate the discussion on page 16 line 8ff.**

We will add the uncontrolled power production, i.e. the power the farm would have produced without the controller, to Figure 7. This is the best comparison to help in the discussion on page 16 line 8 because "total available wind farm power" is difficult to clearly define.

---

## Author Response (AR1)

**Response to Referee 1**

We thank the referee for carefully reading the manuscript. We have included detailed responses to each of the referee remarks/questions below, where the referee comments are in bold face. Line and page numbers refer to the revised marked-up manuscript.

**General comments:**

**This paper presents results of model predictive control of wind farms to provide secondary frequency regulation balancing services for the power grid. A time-varying one-dimensional wake model is presented to model wake advection and wake interactions while trying to enable real-time implementation.**

**Simulations show that the time-varying wake model leads to much better results than the static wake model that is presented and evaluated as a comparison. To fully put the simulation example in context, it would be useful to know the rated power of each of the 84 wind turbines in the example wind farm. Further, what is the rated wind speed for these turbines, what is the mean wind speed of the incoming flow onto the wind farm, and what is the distribution of the incoming wind speeds onto the wind farm in the simulations? What is the turbulence intensity? Or perhaps it is more useful to characterize the turbulence in terms of IEC turbulence characteristics.**

The actuator disk model used to represent the wind turbines in LES assumes an idealized wind turbine operating in region 2 (always operating below the rated wind speed and power). We have added section 5.2 that describes the characteristics on the inlet wind. This includes the mean velocity, standard deviation of the fluctuations, turbulence intensity, and IEC turbulence class. Based on the output of the first row of wind turbines, we have determined the rated power and rated wind speeds as well. We have also added the rotor diameter, hub height, and representative mean and maximum wind speeds to the caption of Figure 3.

**In discussing the results shown on the left side of Figure 5, the authors describe the change in behavior at approximately 5 minutes, though they don't explain why the change in behavior occurs. Can the authors determine a reason for this sudden shift in behavior? It would be useful for readers if the authors also explain the other changes in behavior that are evident, such as around 15 minutes in the upper left 2 plots and around 25 minutes in all of the left plots.**

**Similarly for the right hand plots in Figure 5. The changes in behavior are slower, but there still appear to be qualitative changes in behavior. For instance, in the lower right plot, the behavior before about 29.5 minutes is different from after that time. Can the authors analyze their data further to explain why the change in behavior occurs? And of course similarly for the other right hand plots of Figure 5. And for the plots in Figure 6 as well. By understanding the reasons for the shifts in behavior, both the authors and readers will be able to gain better insight into the properties of the receding horizon control technique used in this paper.**

Thank you for the interesting analysis of the results. We have revised section 6 considerably to better discuss the behavior of the static model. We believe that the controller is switching between two operating states in order to track up and down regulation events. However, because the controller does not include time dynamics, sudden transitions in operating points cause large transient changes in the power production. We see similar qualitative behavior in the dynamic model, but by including time dynamics, the controller can slowly change between operating points and provide improved power tracking.

**Specific comments:**

**1. The second sentence of Section 3.2 does not make sense (Page 9, line 9). It reads: "The row-averaged power and row-averaged, rotor-averaged are defined from velocities $u_{nm}$ measured at every turbine in the wind farm ... " Would the following be more accurate? "The row-averaged power and row-averaged, rotor-averaged downstream wind velocities are defined from velocities $u_{nm}$ measured at every turbine in the wind farm ... "**

Thank you for pointing out this issue. We have changed the sentence to "The row-averaged power and row-and rotor-averaged wind velocities are defined as [Equations], where $u_{nm}$ is the velocity measured at the turbine in the $n$-th row and $m$-th column of the wind farm. "

**2. When discussing Figure 7, the authors repeatedly specifically indicate that these results are based on an LES simulation, while such comments are never mentioned when discussing the results in Figures 5 and 6. Presumably the results in Figures 5 and 6 are also based on LES simulations?**

Yes, Figures 5 and 6 are also based on LES simulations. We have added clarification about the use of LES to test the controllers in the captions of Figures 5 and 6 and at the beginning of section 6.

**Technical corrections:**

Thank you for pointing out these typographical errors.

**1. In equation (7), should the last term in the denominator be divided by D rather than multiplied by D? That is, should the last factor in the denominator be $[1 + 2k_n(x - s_n)/D]^2$ ?**

Corrected.

**2. In the summation in equation (8), should it be $\delta u_m^2$? That is, should the subscript on $\delta u$ be $m$ rather than $n$?**

Corrected.

**3. In the caption for Figure 3, the actuator disk turbine models look to me to be indicated by black "dashes" rather than "lines".**

Corrected.

**4. Page 12, line 17: "form" should be "from"**

Corrected.

**5. Page 13, line 1: "RegA.D4.IC4.TS" should be "RegA.D4.IC3.TS"**

Corrected

**6. Page 13, line 29: "compared to other PJM signal" should be "compared to other PJM signals"**

Corrected.

**7. Page 16 line 6: What are "2/14" and "8/17"? These have no meaning to me.**

The reference to "2/14 and 8/17" was replaced by "the first two historical signals", which is what we are referring to in the text.

**8. References: Please list out each of the authors and do not use "et al." in any of the author lists.**

Corrected.

**Response to Referee 2**

We thank the referee for carefully reading the manuscript. We have included detailed responses to each of the referee remarks/questions below, where the referee comments are in bold face. Line and page numbers refer to the revised marked-up manuscript.

**The paper presents interesting results regarding the ability of wind farms to provide secondary frequency regulation while minimizing the amount of energy not produced. Some points that would improve, in my opinion, the paper:**

**1. The presented wind farm control approach is likely to be computationally too expensive for the use in real wind farms. It would be useful to also discuss how the control approach could be applied to real wind farms.**

In the implementation used in this paper, the optimization step takes approximately 60 seconds on a single processor. While this is 6 times larger than the advancement time of 10 s, several refinements can bring the optimization time to a fraction of the advancement time (allowing real time control). We have added a paragraph at the end of section 3.1 (p. 9 Line 15 of the marked-up manuscript) that discusses how the controlled can achieve real time control.

**2. The introduction could be shortened by moving some of the content to a methodology chapter.**

Thank you for this suggestion. We have moved the paragraph on page 3, line 12 of the marked-up manuscript to Section 7, page 16, line 12 of the marked-up manuscript.

**3. Switching chapter 2 and 3 would improve the flow of the paper.**

We think section 3 requires information from section 2 to be fully coherent and have therefore kept the current organization.

**4. The use of thrust coefficient as input to a wind turbine controller is not realistic.**

We agree using the thrust coefficient as control input is a significant simplification. However, we discuss the future work needed to move toward more realistic controls in the last paragraph of the conclusion (page 22, line 18 of the marked-up manuscript). To help with this discussion, we have added a sentence about the relationship between the thrust coefficient and blade pitch/generator torque on page 7, line 15 of the marked-up manuscript.

**5. Please provide more details regarding the rated power of the wind turbines, their rated wind speed and the mean wind speed considered in the simulations.**

The actuator disk model used to represent the wind turbines in LES assumes an idealized wind turbine operating in region 2 (always operating below the rated wind speed and power). We have added section 5.2 that describes the characteristics on the inlet wind. This includes the mean velocity, standard deviation of the fluctuations, turbulence intensity, and IEC turbulence class. Based on the output of the first row of wind turbines, we have determined the rated power and rated wind speeds as well. We have also added the rotor diameter, hub height, and representative mean and maximum wind speeds to the caption of Figure 3.

**6. Please mention the frequency of the control with regards to the discussion on p. 13 line 13ff**

Control actions are applied every 10 seconds (the receding horizon advancement time). To clarify, we have added a discussion of the frequency of the controller on page 15, line 22 of the marked-up manuscript. The additional paragraph on page 9, line 15 of the marked-up manuscript will also help with this discussion.

**7. Instead of showing the performance of the static model-based controller for all cases it would be useful to focus on a single cases and include figures on rotor effective wind speed at a column of turbines.**

We believe it is useful to show all of the comparison cases to highlight consistent trends in the results. Since the inflow is different between the different cases, we can get a better sense of the overall performance by

looking at a variety of cases. We have added Figure 7 to show more details of row power, rotor effective wind speed, and thrust coefficient for a single simulation case.

**8. The impact of the paper would be improved by a comparison of the performance of the controllers to a standard PI(D) control approach.**

We agree that a comparison to other control designs, particularly those of Aho et al. (2013) and van Wingerden et al. (2017), would be of great interest. This kind of analysis, however, requires a careful comparison of the tracking error, required derate, and possible other financial and market aspects. Since the development of PI(D)-type control techniques (see van Wingerden et al.) is still ongoing (including consideration of the required derate or development a method to distribute set points to individual turbines), we believe a comprehensive comparison is outside the scope of the present paper. Instead, this comparison would be well covered by a collaborative effort among the community. Therefore, we have added this possible comparison as a point for future work at the end of the conclusion.

**9. In chapter 7 please use a quantitative assessment of the controller instead of qualitative statements. It is mentioned that the controller reduces turbulence driven power fluctuations. It would be necessary to justify this state by comparing the controller against a PI(D) control approach.**

In Figure 9 of the marked-up manuscript, we provide a quantitative analysis of the controller using PJM's performance scores. To justify the statement about reduced turbulence driven power fluctuations, we have compared on page 16, line 8 of the marked-up manuscript the variance of the pre-control power about the baseline power $P_{\text{base}}$ to the wind farm power during the controlled period about the reference signal. A comparison to other control approaches, such as PI(D), is outside the scope of this paper.

**10. Please also include the total available wind farm power in the figures. This would also facilitate the discussion on page 16 line 8ff.**

We have added the uncontrolled power production, i.e. the power the farm would have produced without the controller, to Figure 8 of the marked-up manuscript. This is the best comparison to help in the discussion because "total available wind farm power" is difficult to clearly define.

[revised manuscript text omitted]